# Open Source Vizier: Distributed Infrastructure and API for Reliable and Flexible Blackbox Optimization

**Xingyou Song, Sagi Perel, Chansoo Lee, Greg Kochanski, Daniel Golovin**

Google Research, Brain Team

**Abstract**   Vizier is the de-facto blackbox and hyperparameter optimization service across Google, having optimized some of Google's largest products and research efforts. To operate at the scale of tuning thousands of users' critical systems, Google Vizier solved key design challenges in providing multiple different features, while remaining fully fault-tolerant. In this paper, we introduce Open Source (OSS) Vizier, a standalone Python-based interface for blackbox optimization and research, based on the Google-internal Vizier infrastructure and framework. OSS Vizier provides an API capable of defining and solving a wide variety of optimization problems, including multi-metric, early stopping, transfer learning, and conditional search. Furthermore, it is designed to be a distributed system that assures reliability, and allows multiple parallel evaluations of the user's objective function. The flexible RPC-based infrastructure allows users to access OSS Vizier from binaries written in any language. OSS Vizier also provides a back-end ("Pythia") API that gives algorithm authors a way to interface new algorithms with the core OSS Vizier system. OSS Vizier is available at https://github.com/google/vizier.

## 1   Introduction

Blackbox optimization is the task of optimizing an objective function $f$ where the output $f(x)$ is the only available information about the objective. Due to its generality, blackbox optimization has been applied to an extremely broad range of applications, including but not limited to hyperparameter optimization (He et al., 2021), drug discovery (Shields et al., 2021), reinforcement learning (Parker-Holder et al., 2022), and industrial engineering (Zhang et al., 2020).

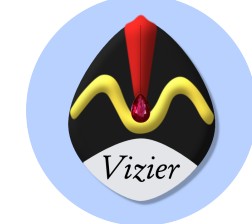

Google Vizier (Golovin et al., 2017) is the first hyperparameter tuning service designed to scale, and has thousands of monthly users both on the research[1] and production side at Google. Since its inception, Google Vizier has run millions of blackbox optimization tasks and saved a significant amount of computing and human resources to Google and its customers.

This paper describes Open Source (OSS) Vizier, a standalone Python implementation of Google Vizier's APIs. It consists of a *user API*, which allows users to configure and optimize their objective

**Figure 1**: Vizier: An advisor.

function, and a *developer API*, which defines abstractions and utilities for implementing new optimization algorithms. Both APIs consist of Remote Procedure Call (RPC) protocols (Section 3) to allow the setup of a scalable, fault-tolerant and customizable blackbox optimization system, and Python libraries (Sections 4.3 and 6) to abstract away the corresponding RPC protocols.

Compared to (Golovin et al., 2017), OSS Vizier features an evolved backend design for algorithm implementations, as well as new functionalities such as conditional search and multi-objective optimization. OSS Vizier's RPC API is based on Vertex Vizier[2], making OSS Vizier compatible with any framework which integrates with Vertex Vizier, such as XManager[3].

---

[1]A list of research works that have used Google Vizier can be found in Appendix C.

[2]https://cloud.google.com/vertex-ai/docs/vizier/overview.   [3]https://github.com/deepmind/xmanager.

Due to the existence of 3 different versions (Google, Vertex/Cloud, OSS) of Vizier, to prevent confusion, we explicitly refer to the version (e.g. "Google Vizier") whenever Vizier is mentioned. We summarize the distinct functionalities of each version of Vizier below:

- Google Vizier: C++ based service hosted on Google's internal servers and integrated deeply with Google's internal infrastructure. The service is available only for Google software engineers and researchers to tune their own objectives with a default algorithm.

- Vertex/Cloud Vizier: C++ based service hosted on Google Cloud servers, available for external customers + businesses to tune their own objectives with a default algorithm.

- OSS Vizier: Fully standalone and customizable code that allows researchers to host a Python-based service on their own servers, for any downstream users to tune their own objectives.

## 2  Problem and Our Contributions

Blackbox optimization has a broad range of applications. Inside Google, these applications include: optimizing existing systems written in a wide variety of programming languages; tuning the hyperparameters of a large ML model using distributed parallel processes (Verbraeken et al., 2020); optimizing a non-computational objective, which can be e.g. physical, chemical, biological, mechanical, or even human-evaluated (Kochanski et al., 2017). Generally, such objectives $f(x)$ we are interested in optimizing possess a moderate number (e.g. several hundred) of parameters for the input $x$, may produce noisy evaluation measurements, and may not be smooth or continuous.

Furthermore, the blackbox optimization workflow greatly varies depending on the application. The evaluation latency can be anywhere between seconds and weeks, while the budget for the number of evaluations, or `Trials`, varies from tens to millions. Evaluations can be done asynchronously (e.g. ML model tuning) or in synchronous batches (e.g. wet lab experiments). Furthermore, evaluations may fail due to transient errors and should be retried, or may fail due to persistent errors (e.g. $f(x)$ cannot be measured) and should not be retried. One may also wish to stop the evaluation process early after observing intermediate measurements (e.g. from a ML model's learning curve) in order to save resources.

To handle all of these scenarios, OSS Vizier is developed as a **service**. The service architecture does not make assumptions on how `Trials` are evaluated, but rather simply specifies a stable API for obtaining suggestions $x_1, x_2, ...$ to evaluate and report results as `Trials`. Users have the freedom to determine when to request trials, how to evaluate trials, and when to report back results.

Another advantage of the service architecture is that it can collect data and metrics over time. Google Vizier runs as a central service, and we track usage patterns to inform our research agenda, and our extensive database of runs serves as a valuable dataset for research into meta-learning and multitask transfer learning. This allows users to transparently benefit from the resulting improvements we make to the system.

### 2.1  Comparisons to Related Work

Table 1 contains a non-comprehensive list of open-source packages for blackbox optimization, focusing on hyperparameter tuning. Overall, OSS Vizier API is compatible with many of the features present in other hyperparameter tuning open-source packages. We did not include commercial services for hyperparameter tuning such as Microsoft Azure, Amazon SageMaker, SigOpt and Vertex Vizier. For a comprehensive review of hyperparameter tuning tools, see (He et al., 2021). There are many other blackbox optimization tools not mentioned in Table 1, including iterated racing (López-Ibáñez et al., 2016; Vieira, 2021), as well as heuristics and automation of algorithm designs (Bezerra et al., 2016; Hoos and Stützle, 2018); see more comparisons and usages in (Lindauer et al., 2022; Feurer et al., 2015).

We divide the open-source packages into three categories:

| Name | Type | Client Languages | Parallel Trials | Features* |
|---|---|---|---|---|
| OSS Vizier | Service | Any | Yes | Multi-Objective, Early Stopping, Transfer Learning, Conditional Search |
| SMAC | Framework | Python | Yes | Multi-Objective, Multi-fidelity, Early Stopping, Conditional Search, Parameter Constraints |
| Advisor | Service | Any | Yes | Early Stopping |
| OpenBox | Service | Any | Yes | Multi-Objective, Early Stopping, Transfer Learning, Parameter Constraints |
| HpBandSter | Framework | Python | Yes | Early Stopping, Conditional Search, Parameter Constraints |
| Ax + BoTorch | Framework | Python | Yes | Multi-Objective, Multi-fidelity, Early Stopping, Transfer Learning, Parameter and Outcome Constraints |
| HyperOpt | Library | Python | No | Conditional Search |
| Emukit | Library | Python | No | Multi-Objective, Multi-fidelity, Outcome Constraints |

**Table 1**: Open Source Optimization Packages. *OSS Vizier supports the API only.

- **Services** host algorithms in a server. OSS Vizier, Advisor (Chen, 2017) and OpenBox (Li et al., 2021), which are modeled after Google Vizier (Golovin et al., 2017), belong to this category. Services are more flexible and scalable than frameworks, at the cost of engineering complexities.

- **Frameworks** execute the entire optimization, including both the suggestion algorithm and user evaluation code. Ax (Facebook, 2021) and HpBandSter (ML4AAD, 2018) belong to this category. While frameworks are convenient, they often require knowledge on the system being optimized, such as how to manage resources and perform proper initialization and shutdown.

- **Libraries** implement blackbox optimization algorithms. HyperOpt (Bergstra et al., 2013), Emukit (Paleyes et al., 2019), and BoTorch (Balandat et al., 2020) belong to this category. Libraries offer the most freedom but lack scalability features such as error recovery and distributed/asynchronous trial evaluations. Instead, libaries are often used as algorithm implementations for frameworks or services (e.g. BoTorch in Ax).

One major architectural difference between OSS Vizier and other services is that OSS Vizier's algorithms may run in a separate service and communicate via RPCs with the API server, which performs database operations. With a distributed backend setup, OSS Vizier can serve algorithms written in different languages, scale up to thousands of concurrent users, and continuously process user requests without interruptions during a server maintenance or update.

Furthermore, there are other minor differences between the services. While OSS Vizier and OpenBox support distinguishing workers via the workers' logical IDs (Section 5), Advisor does not. In addition, OSS Vizier's Python clients possess more sophisticated functionalities than Advisor's, while OpenBox lacks a client implementation and requires users to implement client code using framework-provided worker wrappers. OSS Vizier also emphasizes algorithm development, by providing a developer API called *Pythia* (Section 6) and utility libraries for state recovery. Other features of OSS Vizier include:

- OSS Vizier is one of the first open-source AutoML systems simultaneously compatible with a large-scale industry production service, Vertex Vizier, via our PyVizier library (Section 4.3).

- The backend of OSS Vizier is based on the standard Google Protocol Buffer library, one of the most widely used RPC formats, which allows extensive customizability. In particular, the client (i.e. blackbox function to be tuned) can be written in any language and is not restricted to machine learning models in Python.

- OSS Vizier is extensively integrated with numerous other Google packages, such as Deepmind XManager for experiment management (Section 7).

## 3 Infrastructure

We briefly conceptually define a *Study* as all relevant data pertaining to an entire optimization loop, a *Suggestion* as a suggested $x$, and a *Trial* containing both $x$ and the objective $f(x)$. Note that in the code, we use Trial as a container to store both $x$ and $f(x)$ and thus, a Trial without $f(x)$ is also considered a suggestion. We define these core primitives more programatically in Section 4.

### 3.1 Protocol Buffers

OSS Vizier's APIs are RPC interfaces that carry protocol buffers, or *protobufs/protos*[4], to allow simple and efficient inter-machine communication. The protos are language- and platform- independent objects for serializing structured data, which make building external software layers and wrappers onto the system straightforward. In particular, the user can provide their own:

- **Visualization Tools**: Since OSS Vizier securely stores all study data in its database, the data can then be loaded and visualized, with e.g. standard Python tools (Colab, Numpy, Scipy, Matplotlib) and other statistical packages such as R via RProtoBuf (Eddelbuettel et al., 2014). Front-end languages such as Angular/Javascript may also be used for visualizing studies.

- **Persistent Datastore**: The database in OSS Vizier can changed based on the user's needs. For instance, a SQL-based datastore with full query functionality may be used to store study data.

- **Clients**: Protobufs allow binaries written in Python, C++, and other languages to be tuned and/or used for evaluating the objective function. This allows OSS Vizier to easily tune existing systems.

We explain the interactions between these components in a distributed backend below.

### 3.2 Distributed Backend

In order to serve multiple users while remaining fault-tolerant, OSS Vizier runs in a distributed fashion, with a *server* performing the algorithmic proposal work, while users or *clients* communicate with the server via RPCs using the Client API, built upon gRPC [5]. A packet of RPC communication is formatted in terms of standard Google protobufs.

To start an optimization loop, a client will send a CreateStudy RPC request to the server, and the server will create a new Study in its datastore and return the ID to the client. The main tuning workflow in OSS Vizier will then involve the following repeated cycle of events:

1. The client sends a SuggestTrials RPC request to the server.

2. The server creates a Operation in its datastore, and starts a thread to launch a Pythia policy (i.e. blackbox optimization algorithm) to compute the next suggested Trials. The server returns an Operation protobuf to the client to denote the computation taking place.

3. The client will repeatedly poll the server via GetOperation RPCs to check the status of the Operation until the Operation is done.

4. When the Pythia policy produces its suggestions, the server will store these suggestions into the Operation and mark the Operation done, which will be collected by the client's GetOperation ping.

5. The client retrieves the suggestions $x_i, ..., x_{i+n}$ stored inside the Operation, and returns objective function measurements $f(x_i), ..., f(x_{i+n})$ to the server via calls to the CompleteTrial RPC.

Note that the server may be launched in the same local process as the client, in cases where distributed computing is not needed and functio evaluation is cheap (e.g. benchmarking algorithms

---

[4]https://github.com/protocolbuffers/protobuf  [5]https://grpc.io/

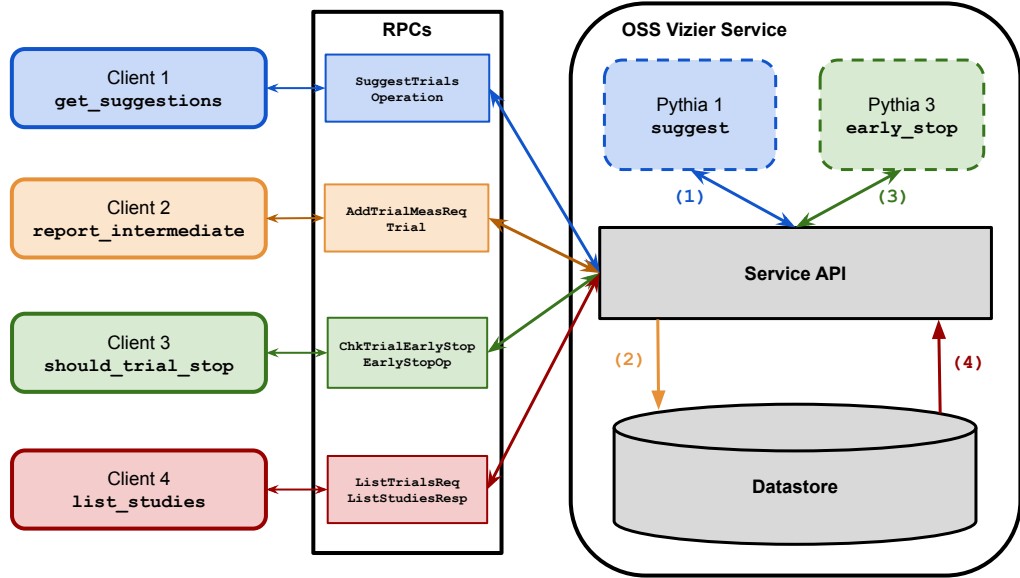

**Figure 2**: Pictorial representation of the distributed pipeline. The OSS Vizier server services multiple clients, each with their own types of requests. Such requests can involve running Pythia Policies, saving measurement data, or retrieving previous studies. Note that Pythia may run as a separate service from the API service.

on synthetic functions). However, if the user wishes to use the distributed setting, the following are core advantages of OSS Vizier's system:

**Server-side Fault Tolerance**. The `Operations` are stored in the database and contain sufficient information to restart the computation after a server crash, reboot, or update.

**Automated/Early Stopping**. A similar sequence of events takes place when the client sends a `CheckTrialEarlyStoppingStateRequest` RPC, in which the policy determines if a trial's evaluation should be stopped, and returns this signal as a boolean via the `EarlyStoppingOperation` RPC.

**Batched/Parallel Evaluations**. Note that *multiple clients may work on the same study, and the same* `Trial`. This is important for compute-heavy experiments (e.g. neural architecture search) which need to parallelize workload by using multiple machines, with each machine $j$ evaluating the objective $f(x_j)$ after being given suggestion $x_j$ from the server.

**Client-side Fault Tolerance**. When one of the parallel workers fails and then reboots, the service will assign the worker the same suggestion as before. The worker can choose to load a model from the checkpoint to warm-start the evaluation.

## 4 Core Primitives

In Figure 3, we provide a pictorial example representation of how OSS Vizier's primitives are structured; below we provide definitions.

### 4.1 Definitions

A `Study` is a single optimization run over a feasible space. Each study contains a name, its description, its state (e.g. `ACTIVE`, `INACTIVE`, or `COMPLETED`), a `StudySpec`, and a list of suggestions and evaluations (`Trials`).

A `StudySpec` contains the configuration details for the `Study`, namely the search space $\mathcal{X}$ (constructed by `ParameterSpecs`; see §4.2), the algorithm to be used, automated stopping type (see

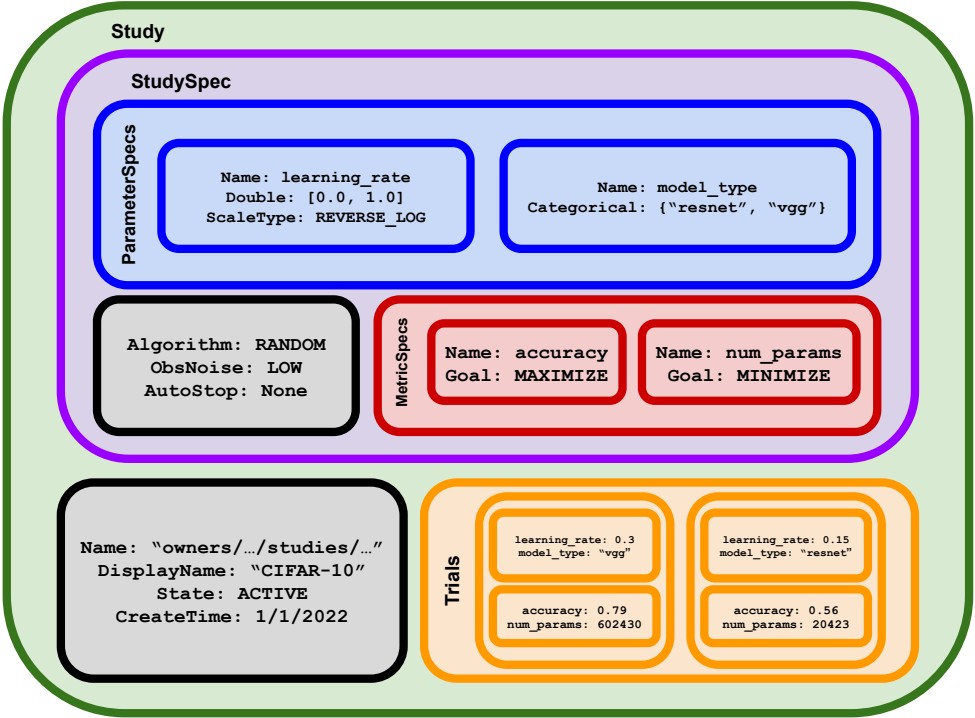

**Figure 3:** Example of a study that tunes a deep learning task, featuring relevant data types.

Appendix B.1), the type of ObservationNoise (see Appendix B.2), and at least one MetricSpec, containing information about the metric $f$ to optimize, including the metric name and the goal (i.e. whether to minimize or maximize $f$). Multiple MetricSpecs will be used for cases involving multiobjective optimization, where the goal is to find Pareto frontiers over multiple objectives $f_1, ..., f_k$.

A Trial is a container for the input $x \in \mathcal{X}$, as well as potentially the scalar value $f(x)$ or multiobjective values $f_1(x), ..., f_k(x)$. Each Trial possesses a State, which indicates what stage of the optimization process the Trial is in, with the two primary states being ACTIVE (meaning that $x$ has been suggested but not yet evaluated) and COMPLETED (meaning that evaluation is finished, and typically that the objectives $(f_1(x), ..., f_k(x))$ have been calculated).

Both the StudySpec and the Trials can contain Metadata. Metadata is not interpreted by OSS Vizier, but rather a convenient method for developers to store algorithm state, by users to store small amounts of arbitrary data, or as an extra communication medium between user code and algorithms.

## 4.2 Search Space

Search spaces can be built by combining the the following primitives, or ParameterSpecs:

- Double: Specifies a continuous range of possible values in the closed interval $[a, b]$ for some real values $a \leq b$.

- Integer: Specifies an integer range of possible values in $[a, b] \in \mathbb{Z}$ for some integers $a \leq b$.

- Discrete: Specifies a finite, ordered set of values from $\mathbb{R}$.

- Categorical: Specifies an unordered list of strings.

Furthermore, each of the numerical parameters {`Double`, `Integer`, `Discrete`} has a *scaling type*, which toggles whether the underlying algorithm is performing optimization in a transformed space. The scale type allows the user to conveniently inform the optimizer about the shape of the function, and can sometimes drastically accelerate the optimization. For instance, a user may use logarithmic scaling, which expresses the intent that a parameter ranging over $[0.001, 10]$ should roughly receive the same amount of attention in the subrange $[0.001, 0.01]$ as $[1, 10]$, which would otherwise not be the case when using uniform scaling.

Each parameter also can potentially contain a list of child parameters, each of which will be active only if the parent's value matches the correct value(s). This allows the notion of *conditional search*, which is helpful when dealing with search spaces involving incompatible parameters or parameters which only exist in specific scenarios. For example, this can be useful when competitively tuning several machine learning algorithms along with each algorithm's parameters. E.g. one could tune the following for the `model` parameter: {`"linear"`, `"DNN"`, `"random_forest"`}, each with its own set of parameters. Conditional parameters help keep the user's code organized, and also describe certain invariances to OSS Vizier, namely that when `model="DNN"`, $f(x)$ will be independent of the `"random_forest"` and `"linear"` model parameters.

These parameter primitives can be used flexibly to build highly complex search spaces, of which we provide examples in Appendix A.

### 4.3 PyVizier

All the above objects are implemented as protos to allow RPC exchanges through the service, as mentioned in Section 3. However, for ease-of-access, each object is also represented by an equivalent *PyVizier* class to provide a more Pythonic interface, validation, and convenient construction (further details and examples are found in Appendix D.3). Translations to and from protos are provided by the `to_proto()` and `from_proto()` methods in PyVizier classes. **PyVizier provides a common interface across all Vizier variants (i.e. Google Vizier, Vertex Vizier, and OSS Vizier)**[6]. The two intended primary use cases for PyVizier are:

- Tuning user binaries. For such cases, the core PyVizier primitive is the `VizierClient` class that allows communication with the service.

- Developing algorithms for researchers. In this case, the core PyVizier primitives are the Pythia `Policy` and `PolicySupporter` classes.

Both cases typically use the `StudyConfig` and `SearchSpace` classes to define the optimization, and the `Trial`, and `Measurement` classes to support the evaluation. We describe the two cases in detail below.

## 5  User API: Parallel Distributed Tuning with OSS Vizier Client

The OSS Vizier service must be set up first (see pseudocode in Appendix D.2), preferably on a multithreaded machine capable of processing multiple RPCs concurrently. Then, replicas of Code Block 1 can be launched in parallel, each with a unique command-line argument to be used as the client id in Line 11. The first replica to be launched creates a new `Study` from the `StudyConfig`, which defines the search space, relevant metrics to be evaluated, and the algorithm for providing suggestions. The other replicas then load the same study to be worked on. There are a few important aspects worth noting in this setting:

- The service does not make any assumptions about how `Trials` are evaluated. Users may complete `Trials` at any latency, and may do so with a custom client written in any language. Algorithms

---

[6]For compatibility reasons, protos have slightly different names than PyVizier equivalents; e.g. `StudySpec` protos are equivalent to `StudyConfig` PyVizier objects. We describe conversions further in Appendix D.3

```python
from vizier import StudyConfig, VizierClient

config = StudyConfig() # Search space, metrics, and algorithm.
root = config.search_space.select_root() # "Root" params must exist in every trial.
root.add_float('learning_rate', min=1e-4, max=1e-2, scale='LOG')
root.add_int('num_layers', min=1, max=5)
config.metrics.add('accuracy', goal='MAXIMIZE', min=0.0, max=1.0)
config.algorithm = 'RANDOM_SEARCH'

client = VizierClient.load_or_create_study(
    'cifar10', config, client_id=sys.argv[1]) # Each client should use a unique id.

while suggestions := client.get_suggestions(count=1)
  # Evaluate the suggestion(s) and report the results to Vizier.
  for trial in suggestions:
    metrics = _evaluate_trial(trial.parameters)
    client.complete_trial(metrics, trial_id=trial.id)
```

**Code Block 1**: Pseudocode for tuning a blackbox function using the included Python client. To save space, we did not use longer official argument names from the actual code.

may however, set a time limit and reassign `Trials` to other clients to prevent stalling (e.g. due to a slow client).

- Each `Trial` is assigned a `client_id` and only suggested to clients created with the same `client_id`. This design makes it easy for users to recover from failures during `Trial` evaluations; if one of the tuning binaries is accidentally shut down, users can simply restart the binary with the same client id. The tuning binary creates a new client attached to the same study and OSS Vizier suggests the same `Trial`.

- Multiple binaries can share the same `client_id` and collaborate on evaluating the same `Trial`. This feature is useful in tuning a large distributed model with multiple workers and evaluators.

- The client may optionally turn on automated stopping for objectives that can provide intermediate measurements (e.g. learning curves in deep learning applications). Further details and an example code snippet can be found in Appendix B.1 and Appendix 3 respectively.

## 6 Developer API: Implementing a New Algorithm Using Pythia Policy

### 6.1 Overview

As we have explained in Section 3, OSS Vizier runs its algorithms in a binary called the *Pythia service* (which can be the same binary as the API service). When the client asks for suggestions or early stopping decisions, the API service creates operations and sends requests to the Pythia service. This section describes the default python implementation of the Pythia service included in the open-source package.

The Pythia service creates a `Policy` object that executes the algorithm and returns the response. `Policy` is designed to be a minimal and general-purposed interface built on top of PyVizier, to allow researchers to quickly incorporate their own blackbox optimization algorithms. `Policy` is usually given a `PolicySupporter`, which is a mini-client specialized in reading and filtering `Trials`. As shown in Code Block 2, a typical `Policy` loads `Trials` via `PolicySupporter` and processes the request at hand.

```
1  from vizier.pythia import Policy, PolicySupporter, SuggestRequest, SuggestDecisions
2
3  class MyPolicy(Policy):
4    def __init__(self, policy_supporter: PolicySupporter):
5      self.policy_supporter = policy_supporter  # Used to obtain old trials.
6
7    def suggest(self, request: SuggestRequest) -> SuggestDecisions:
8      """Suggests trials to be evaluated."""
9      Xs, y = _trials_to_np_arrays(self.policy_supporter.GetTrials(
10         status='COMPLETED')) # Use COMPLETED trials only.
11     model = _train_gp(Xs, y)
12     return _optimize_ei(model, request.study_config.search_space)
```

**Code Block 2**: Pseudocode for implementing a Gaussian Process Bandit.

### 6.2 PolicySupporter

The PolicySupporter allows the Policy to actively decide what Trials from what Studies are needed to generate the next batch of Suggestions. Policies can meta-learn from potentially any Study in the database by calling the GetStudyConfig and GetTrials methods. Beyond that, the Policy can request only the Trials it needs; e.g. for algorithms that only need to look at newly evaluated Trials, this can reduce the database work by orders of magnitude relative to loading all the Trials.

### 6.3 State Saving via Metadata

The primary application of Google Vizier (Golovin et al., 2017) was optimizing a blackbox function that is expensive to evaluate. Over time, as Google Vizier became widely adopted, there was an increasing number of applications where users wished to evaluate cheap functions over a very large number of Trials. Popular methods for these applications include evolutionary methods and local search methods, such as NSGA-II (Deb et al., 2002), Firefly (Yang, 2010), and Harmony Search (Lee and Geem, 2005) to name a few (For a survey on meta-heuristics, see Beheshti and Shamsuddin (2013)).

A typical algorithm in this category iteratively updates its population pool and generates mutations to be suggested, both of which take constant time with respect to the number of previous trials, as opposed to e.g. cubic time when using Gaussian Processes in a Bayesian Optimization loop. Since the lifespan of a Policy object is equivalent to that of one suggestion or early stopping operation, the algorithm would need to fetch all Trials in the Study and reconstruct its state in $O$(number of previous trials) time. This leads to slow and difficult-to-maintain implementations.

PolicySupporter provides an easy-to-use API for developers to send algorithm states into the database as Metadata. Metadata is a key-value mapping with namespaces that help prevent key collisions. There are two tables for metadata in the database: one attached to the StudySpec and another to each Trial. A Policy can restore its last saved state from metadata, reflect the recently added Trials, and process the operation at hand. We provide example code for this functionality in Appendix D.4

## 7 Integrations

OSS Vizier is also compatible with multiple other interfaces developed at Google as well. These include:

---

[7]https://cloud.google.com/vertex-ai/docs/reference/rest/v1beta1/StudySpec.

- Vertex Vizier whose Protocol Buffer definitions are exactly the same[7] as OSS Vizier's. This consistency also allows a wide variety of other packages (discussed below) pre-integrated with Vertex Vizier to be used with minimal changes.

- Deepmind XManager experiments currently can be tuned by Vertex Vizier[8] through `VizierWorker`. This worker can also be directly connected to an OSS Vizier server to allow custom policies to manage experiments.

- OSS Vizier will also be the core backend for PyGlove (Peng et al., 2020)[9], which is a symbolic programming language for AutoML, in particular facilitating combinational and evolutionary optimization which are common in neural architecture search applications.

## 8  Conclusion, Limitations and Broader Impact Statement

**Conclusion**. We discussed the motivations and benefits behind providing OSS Vizier as a service in comparison to other blackbox optimization libraries, and described how our gRPC-based distributed back-end infrastructure may be deployed as a fault-tolerant yet flexible system that is capable of supporting multiple clients and diverse use cases. We further outlined our client-server API for tuning, our algorithm development Pythia API, and integrations with other Google libraries.

**Limitations**. Due to proprietary and legal concerns, we are unable to open-source the default algorithms used in Google Vizier and Cloud Vizier. Furthermore, this paper intentionally does not discuss algorithms or benchmarks, as the emphasis is on the systems aspect of AutoML. Algorithms may easily be added as policies to OSS Vizier's collection over time from contributors.

OSS Vizier also may not be suitable for all problems within the very broad scope of blackbox optimization. For instance, if evaluating $f(x)$ is very cheap and fast (e.g. miliseconds), then the OSS Vizier service itself may dominate the overall cost and speed. Furthermore, for problems requiring very large numbers of parameters (e.g. 100K+) and evaluations (e.g. 1M+), such as training a large neural network with gradientless methods (Mania et al., 2018; Such et al., 2017), OSS Vizier can also be inappropriate, as such cases can overload the datastore memory with redundant trials which do not need to be kept track of.

**Broader Impact**. While there are a rich collection of sophisticated and effective AutoML algorithms published every year, broad adoption to practical use cases still remains low, as only 7% of the ICLR 2020 and NeurIPS 2019 papers used a tuning method other than random or grid search (Bouthillier and Varoquaux, 2020). In comparison, Google Vizier is widely used among multiple researchers at Google, including for conference submissions. We hope that the release of OSS Vizier and its similar benefits may significantly improve the reach of AutoML techniques to users.

In terms of potential negative impacts, optimization as a service encourages central storage of data with the attendant risks and benefits. For example, currently through the Client API, a user may request all studies associated with another users, which may cause security and privacy concerns. This may be fixed by limiting user access to only their own studies in the service logic. Furthermore, the host of the service currently has full access to all client data, which is another potential privacy concern. However, from our experience with Google Vizier, the most impactful applications for clients typically occur when parameters and measurements correspond to aggregate data (e.g. the learning rate of a ML algorithm, or e.g. the number of threads in a server) rather than data that describes individuals. Furthermore, data received by OSS Vizier can be obscured to a degree to reduce unwanted exposure to the host. Most notably, names (e.g. study name, parameter and metric names) can be encrypted, and (within limits) differential privacy (Dwork, 2008) approaches, especially for databases (Johnson et al., 2018), can be applied to the parameters values and measurements.

---

[8]https://github.com/deepmind/xmanager/tree/main/xmanager/vizier.   [9]PyGlove will be open-sourced soon.

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
