# Appendix

## A Search Space Flexibility

In this section, we describe the ways in which more complex search spaces may be created in OSS Vizier, showcasing its flexibility and applicability to a wide variety of problems.

### A.1 Combinatorial Optimization

One of the most common uses for blackbox optimization in research involves combinatorial optimization. In this setting, $\mathcal{X}$ is usually defined via common manipulations over the set $[n] = \{0, 1, ..., n-1\}$, such as permutations or subset selections. Below, we provide example methods to deal with such cases, in the order of most practical to least practical. We note that many of these methods are more suited for evolutionary algorithms which only need to utilize mutations and cross-overs between trials, rather than regression-based methods (e.g. Bayesian Optimization).

**A.1.1 Reparameterization**. Reparameterization of the search space $\mathcal{X}$ via conceptual means should be considered first, as it is one of the most practical and easiest ways to reduce the complexity of representing $\mathcal{X}$ in OSS Vizier. Mathematically speaking, the high level idea is to construct a more practical search space $\mathcal{Z}$ which can easily be represented in OSS Vizier, and then create a surjective mapping $\Phi : Z \to X$.

For basic combinatorial objects such as permutations, if we consider the standard permutation space $\mathcal{X} = \{x : x \in [n]^n, x_i \neq x_j \ \forall i \neq j\}$, then we may define $\mathcal{Z} = [n] \times [n-1] \times ... \times [2] \times [1]$ and allow $\Phi$ to be the decoding operator for the Lehmer code[10]. If $\mathcal{X} = \{x : x \subseteq [n], |x| = k\}$ involves subset selection, then we may define $\mathcal{Z} = [n] \times [n-1] \times ... \times [n-k+1]$ and apply a similar mapping.

Another common case involves searching over the space of graphs. In such scenarios, there are a multitude of methods to parameterizing the graph, including adjacency matrices via $[n] \times [n]$. An illustrative example can be seen across neural architecture search (NAS) benchmarks. Even though such search spaces correspond to graph objects, ironically, many NAS benchmarks, termed "NASBENCH"s, actually do not use nested or conditional search spaces. For instance, NASBENCH-101 (Ying et al., 2019) uses only a flat adjacency matrix and flat operation list. NASBENCH-201 (Dong and Yang, 2020) is even simpler, as it takes the graph dual of the node-op representation, allowing the search space to be a full feasible set represented by only 5 categorical parameters.

**A.1.2 Infeasibility**. In some scenarios, we may not be able to find a mapping $\Phi$ as in the reparameterization case above, but instead may lift the search space $\mathcal{X}$ into a larger search space $\mathcal{Z}$, where $\mathcal{X} \subset \mathcal{Z}$, and thus perform search on $\mathcal{Z}$ instead. For trials in $\mathcal{Z} - \mathcal{X} = \{z : z \in \mathcal{Z}, z \notin X\}$, OSS Vizier supports reporting these trials as infeasible. As a basic example, if $\mathcal{X} = \{x \in \mathbb{R}^2 : ||x|| \leq 1\}$ defines a disk, then $\mathcal{Z} = [-1, 1]^2$. Another example can be seen with the same NASBENCH-101 (Ying et al., 2019) benchmark described earlier, where some pairs of adjacency matrices and operation lists do not correspond to an actual valid graph, and are thus infeasible.

The main limitation is if $|\mathcal{X}| \ll |\mathcal{Z}|$, the vast bulk of trials may be infeasible, and if so, the search will converge slowly. Furthermore, for the disk case, this can lead to problems during optimization, as it creates a sharp border $\mathcal{X} \cap \mathcal{Z}$ and a flat infeasible region $\mathcal{Z} - \mathcal{X}$. This leads to lack of information about which infeasible points are better/worse than others, and can also make it difficult to find a small feasible region. Modelling techniques such Gaussian Processes also inherently assume the objective function is continuous everywhere, which is incompatible with the discontinuity from the border $\mathcal{X} \cap \mathcal{Z}$.

---

[10] https://en.wikipedia.org/wiki/Lehmer_code

**A.1.3** **Serialization.** If all else fails, we may avoid the use of the *ParameterSpec* API and simply serialize $x \in \mathcal{X}$ into a string format, which can then be inserted into a Trial's *metadata* field. In cooperation with a custom Pythia policy, this can be very effective.

## B  Additional OSS Vizier Settings

### B.1  Automated Stopping

Automated/early stopping is used commonly when trials can be stopped early to save resources, and is determined by the trial's intermediate measurements. Currently there are two modes to automated stopping which the client can specify in their `StudyConfig`:

- Decay Curve Automated Stopping, in which a Gaussian Process Regressor is built to predict the final objective value of a Trial based on the already completed Trials and the intermediate measurements of the current Trial. Early stopping is requested for the current Trial if there is very low probability to exceed the optimal value found so far.

- Median Automated Stopping, in which a pending trial is stopped if the Trial's best objective value is strictly below the median 'performance' of all completed Trials reported up to the Trial's last measurement. Currently, 'performance' refers to the running average of the objective values reported by the Trial in each measurement.

### B.2  Observation Noise

We have found it useful to let the user give Vizer a hint about the amount of noise in their evaluations via the `StudyConfig`. Because the noise/irreproducibility of evaluations is often not well known in advance by users, we give users a broad choice that the noise is either `Low` or `High`:

- `Low`: This implies that the objective function is (nearly) perfectly reproducible, and an algorithm should never repeat the same Trial parameters.

- `High`: This assumes there is enough noise in the evaluations that it is worthwhile for OSS Vizier sometimes to re-evaluate with the same (or nearly) parameter values.

This hint is passed to the Pythia policy, and the policy is free to also use this hint to e.g. adjust priors on the hyperparameters of a Gaussian Process regressor.

## C  Google Vizier Users and Citations

Besides Google Vizier's extensive internal production usage, below comprises a selected list of publicly available research works[11] which have used Google Vizier, demonstrating its rich research user-base which may directly translate to OSS Vizier's future user-base as well.

**Neural Architecture Search.** Google Vizier has acted as a core backend for many of the neural architecture search (NAS) efforts at Google, beginning with Google Vizier having been used to hyperparameter tune the RNN controller in the original NAS paper (Zoph and Le, 2017). Over the course of NAS research, Google Vizier has also been used to reliably handle the training of thousands of models (Zoph et al., 2018; Chen et al., 2018), as well as comparisons against different NAS optimization algorithms in NASBENCH-101 (Ying et al., 2019). Furthermore, it serves as the primary distributed backend for PyGlove (Peng et al., 2020), a core evolutionary algorithm API for NAS research across Google.

**Hardware and Systems.** Google Vizier's tuning led to crucial gains for hardware benchmarking, such as improving JAX's MLPerf scores over TPUs [12]. Google Vizier's multiobjective optimization capabilities were a key component in producing better computer architecture designs in APOLLO (Yazdanbakhsh et al., 2020) [13]. Furthermore, Google Vizier was a key component to *Full-stack Accelerator Search Technique* (FAST) (Zhang et al., 2022), an automated framework for jointly optimizing hardware datapath, software schedule, and compiler passes.

**Reinforcement Learning.** "AutoRL" (Parker-Holder et al., 2022) has recently seen a great deal of promise in automating reinforcement learning systems. Google Vizier was extensively used as the core component in tuning hyperparameters and rewards in navigation (Faust et al., 2019; Francis et al., 2020; Chiang et al., 2019). Google Vizier's backend was also used to host the Regularized Evolution optimizer (Real et al., 2019), used for evolving RL algorithms (Co-Reyes et al., 2021), where the search space involved combinatorial directed acyclic graphs (DAGs). On the infrastructure side, Google Vizier was used to improve the performance of Reverb (Cassirer et al., 2021), one of the core replay buffer APIs used for most RL projects at Google. (Agarwal et al., 2021)

**Biology/Chemistry/Healthcare.** Google Vizier's algorithms were used for comparison on several papers related to protein optimization (Bileschi et al., 2022), and was also used to tune RNNs for peptide identification in (Tiwary et al., 2019). For healthcare, Google Vizier was used to tune models for classifying diseases such as diabetic retinopathy (Krause et al., 2017)

**General Deep Learning.** For fundamental research, Google Vizier was used to tune Neural Additive Models (Agarwal et al., 2020), and has also been the backbone of core research into infinite-width deep neural networks, having tuned (Nguyen et al., 2021; Lee et al., 2020; Hron et al., 2020b,a). For NLP-based tasks, Google Vizier regularly tunes language model training, and has also been used to search feature weights in (Wang et al., 2020), as well improve performance for work on theorem proving (Aygün et al., 2020). Computer vision models such as ones used for the Pixel-3[14] have been tuned by Google Vizier.

**Miscallaneous:.** As an example of tuning for human-based judgement on objectives unrelated to technology, Google Vizier was used to tune the recipe for cookie-baking (Kochanski et al., 2017).

---

[11] Full list of Google Vizier's citations: https://scholar.google.com/scholar?oi=bibs&hl=en&cites=14342343058535677299.

[12] Link too long; hyperlink can be found here.

[13] https://ai.googleblog.com/2021/02/machine-learning-for-computer.html

[14] https://ai.googleblog.com/2018/12/top-shot-on-pixel-3.html

## D Extended Code Samples

### D.1 Automated stopping

Code Block 3 demonstrates the use of automated stopping, when training a standard machine learning model.

```python
from vizier import StudyConfig, VizierClient

config = StudyConfig()
... # configure search space and metrics
client = VizierClient.load_or_create_study(
    'cifar10', config, client_id=sys.argv[1]) # Each client should use a unique id.

while suggestions := client.get_suggestions(count=1)
  # Evaluate the suggestion(s) and report the results to OSS Vizier.
  for trial in suggestions:
    for epoch in range(EPOCHS):
      if client.should_trial_stop(trial.id):
        break
      metrics = model.train_and_evaluate(trial.parameters['learning_rate'])
      client.report_metrics(epoch, metrics)
    metrics = model.evaluate()
    client.complete_trial(metrics, trial_id=trial.id)
```

Code Block 3: *Pseudocode* for tuning a model using the included Python client, with early stopping enabled.

### D.2 Service Setup

Code Block 4 displays the simple method in which to setup the service on a multithreaded server.

```python
from vizier.service import vizier_server
from vizier.service import vizier_service_pb2_grpc

hostname = 'localhost' # Example; usually user-specified
port = 6006 # Example; usually user-specified
address = f'{hostname}:{port}'
servicer = vizier_server.VizierService()

server = grpc.server(futures.ThreadPoolExecutor(max_workers=100))
vizier_service_pb2_grpc.add_VizierServiceServicer_to_server(servicer, server)
server.add_secure_port(address, grpc.local_server_credentials())
server.start()
```

Code Block 4: Pseudocode for setting up the service on a server.

### D.3 Proto vs Python API

We provide an example of equivalent methods between PyVizier and corresponding Protocol Buffer objects. Note that clients and algorithm developers should not normally need to modify protos. Such cases are more common if one wishes to add extra layers on top of the service, as mentioned in Subsection 3.1.

```python
from vizier.service import study_pb2
from google.protobuf import struct_pb2

param_1 = study_pb2.Trial.Parameter(parameter_id='learning_rate', value=struct_pb2.
    Value(number_value=0.4))
param_2 = study_pb2.Trial.Parameter(parameter_id='model_type', value=struct_pb2.
    Value(string_value='vgg'))
metric_1 = study_pb2.Measurement.Metric(metric_id='accuracy',value=0.4)
metric_2 = study_pb2.Measurement.Metric(metric_id='num_params',value=20423)
final_measurement = study_pb2.Trial.Measurement(metrics=[metric_1,metric_2])
trial = study_pb2.Trial(parameters=[param_1,param_2], final_measurement=
    final_measurement)
```

Code Block 5: Original Protocol Buffer method of creating a Trial.

```python
from vizier.pyvizier import ParameterDict, ParameterValue, Measurement, Metric,
    Trial

params=ParameterDict()
params['learning_rate'] = ParameterValue(0.4)
params['model_type'] = ParameterValue('vgg')
final_measurement = Measurement()
final_measurement.metrics['accuracy'] = Metric(0.7)
final_measurement.metrics['num_params'] = Metric(20423)
trial = pv.Trial(parameters=params,final_measurement=final_measurement)
```

Code Block 6: Equivalent method of writing the PyVizier version of the Trial from Code Block 5. Note the significantly more "Pythonic" way of writing code, with a significant reduction in code complexity.

We also provide in Table 2, changes between OSS Vizier's Protocol Buffer names and their corresponding PyVizier names, as well as converter objects.

| Protocol Buffer Name | PyVizier Name | Converter |
|---|---|---|
| Study | Study | N/A |
| StudySpec | SearchSpace + StudyConfig | SearchSpace (self) + StudyConfig (self) |
| ParameterSpec | ParameterConfig | ParameterConfigConverter |
| Trial | Trial | TrialConverter |
| Parameter | ParameterValue | ParameterValueConvereter |
| MetricSpec | MetricInformation | MetricInformation (self) |
| Measurement | Measurement | MeasurementConverter |

Table 2: Corresponding names and conversion objects between Protocol Buffer and PyVizier objects. (self) denotes that the PyVizier object has its own immediate to_proto() and from_proto() functions.

### D.4 Implementing an Evolutionary Algorithm

OSS Vizier possesses an abstraction `SerializableDesigner` defined purely in terms of PyVizier without any Pythia dependencies. This interface wraps around most commonly used algorithms which sequentially update their internal states as new observations arrive. The interface is easy to understand and can be wrapped into a Pythia policy using the `SerializableDesignerPolicy` class which handles state management. See Code Block 7 for an example.

```python
from vizier import pyvizier as vz

class RegEvo(SerializableDesigner):

  # override
  def suggest(self, count: Optional[int]) -> Sequence[vz.TrialSuggestion]
    """Generate `count` number of mutations and return them."""

  # override
  def update(self, delta: CompletedTrials):
    """Apply selection step and update the population pool."""

  # override
  def dump(self) -> vz.Metadata:
    """Dumps the population pool."""
    md = vz.Metadata()
    md['population'] = json.dumps(...)
    return md

  # override
  def recover(cls: Type['_S'], metadata: vz.Metadata) -> '_S':
    """Restores the population pool."""
    if 'population' not in md:
      raise HarmlessDecodeError('Cannot find key: "population"')
    ... = json.loads(md['population'])

policy = SerializableDesignerPolicy(
    policy_supporter,
    designer_factory=RegEvo.__init__,
    designer_cls=RegEvo)
```

Code Block 7: Example Pseudocode of implementing an evolutionary algorithm as a Pythia policy using `SerializableDesigner` interface.