# OpenReview forum: "Open Source Vizier: Distributed Infrastructure and API for Reliable and Flexible Blackbox Optimization"
_automl.cc/AutoML/2022/Track/Main — AutoML-Conf 2022 (Main Track)_

### Official Review · Reviewer_yaum · 2022-03-31

**Potential Impact On The Field Of Automl Rating:** 3
**Technical Quality And Correctness Rating:** 3
**Clarity Rating:** 3

**Summary Of Contributions:**

This paper introduces Open Source (OSS) Vizier, an open-source package for black box optimization, based on Google Vizier. OSS Vizier provides a distributed system and service API for reliable and flexible optimization. The RPC-based infrastructure allows the users to implement client code using any language. The developer API allows the users to implement new algorithms conveniently.

**Clarity:**

It would be better to add simple implement introduction about the functionalities like multi-objective early stopping and so on.

**Overall Review:**

Positive:
1.	OSS Vizier provides a reliable optimization by the distributed system and a flexible optimization by service API.
2.	OSS Vizier uses the RPCs to communicate which is compatible with other framework, like XManager.

Negative:
1.	OSS Vizier lacks more examples about how to utilize the functionalities and some experiment results on the simple synthetic functions.

**Potential Impact On The Field Of Automl:**

The OSS Vizier provides a fault-tolerant service for black box optimization, and it has many functionalities, e.g., multi-objective, early stopping, transfer learning, conditional search. Users can use it to optimize their own black box problem conveniently, so I hold a positive option on this package. It would be better to provide more examples about how to use the above functionalities.

**Reproducibility:**

The implement code is provided, and I believe researchers can reproduce the results.

**Review Confidence:**

4: You are confident in your assessment, but not absolutely certain. It is unlikely, but not impossible, that you did not understand some parts of the submission or that you are unfamiliar with some pieces of related work.

**Review Rating:**

5: Accept, good paper

**Review Summary:**

This paper introduces OSS Vizier, an open-source package for reliable and flexible black box optimization. Researchers can use this package to solve their problems more quickly and easily. Therefore, I think it can be accepted.

**Technical Quality And Correctness:**

The proposed OSS Vizier introduces the implement details about the package, and provides a full code implement. Though the default algorithms and benchmarks used in Google Vizier cannot be released due to proprietary and legal concerns, some experiments on simple synthetic functions are expected.

---

### Official Review · Reviewer_n6CE · 2022-04-04

**Potential Impact On The Field Of Automl Rating:** 2
**Technical Quality And Correctness Rating:** 3
**Clarity Rating:** 3
**Ethics Rating:** Yes, privacy / security / safety

**Summary Of Contributions:**

OSS Vizier provides a standalone - as a service -  API for blackbox optimisation with a focus on hyperparameter optimisation. Further, an API for users to interface their own algorithms with OSS Vizier enables the use of those too. The main difference between (OSS) Vizier and other frameworks is the distributed design, allowing function evaluations to be distributed over multiple clients that each report back the objective performance to the server running the optimisation algorithm.

**Clarity:**

While indicating a focus on hyperparameter optimisation (HPO), the paper also discusses general black-box optimisation (BBO). As a result, the goal of OSS Vizier is not very crisply defined, perhaps because there is an aim to be general (or possibly in the future). Clearly there is overlap between the two, but depending on the problem a method for one may not be good for the other (or even usable).

An issue is that the connection between the many (apparently connected/related) tools makes it somewhat difficult to follow how things relate. At the same time, when someone is familiar with these other tools, I think this is of interest, too. Related issues/suggestions:
- Perhaps a version of Fig. 3 that indicates which parts are provided by OSS Vizier, and what is provided by other tools (Vizier, PyVizier, Vertex Vizier, ...) would resolve this. Note that the current 'Vizier Service' box does not do this sufficiently; since OSS Vizier provides a server as well, I expect at least part of this box to be provided by OSS Vizier. Some things may be provided by multiple tools, but since this paper is about OSS Vizier, it should be clear at least what that provides (possibly indicate all tools that can provide a certain component). Although I understand mentioning connections and interoperability with other tools is interesting and potentially useful (although it is also unclear to me how the usefulness extends to outside Google), currently I think it causes (me) more confusion than that it helps. Perhaps first making clear (with a figure?) what the relation between all these things is, and then sticking to explaining OSS Vizier and OSS Vizier only is better.

"Multiple binaries can share the same client_id and" --> It is not clear to me how this combines with each client having a unique idea. Do clients run multiple binaries in parallel (in addition to having multiple parallel clients)? Clarification would be helpful.

Minor issues/typos:
- Sect. 3.1: "The database in Vizier can changed based on" --> ...can be changed...
- Sect. 6.3: "evolutionary algorithms, which have seen a surge in popularity since Google Vizier (Golovin et al., 2017) was first introduced." --> This seems highly questionable to me. Unless you can back it up (and do so), the claim that EAs have grown in popularity following and due to the release of Vizier should be left out.
- Sect. 8: "Furthermore, this paper intentionally does not discuss algorithms or benchmarks, as the emphasis is on the systems aspect of AutoML." --> This sentence is strange, and I think the authors probably did not write what they intended (or did so in a way that confuses me). I don't doubt the system is potentially useful (including in access to AutoML), but I don't see what is AutoML about the system itself (besides the algorithms, which the authors exclude from discussion).

**Ethics Details (Optional):**

OSS Vizier collects data on the optimisation problems submitted by users, revealing this to the server owner/provider. The collection of data in general is cause for concern because of privacy reasons, but the authors discuss some options for how this might be mitigated.

Even so, I think the discussion of 'negative impacts' in the conclusion is somewhat limited. Particularly when releasing a tool to make HPO/BBO more accessible, implying the user should know to only expose aggregate data is not reasonable (the more accessible, the less you can expect from your user). Further, that data can be obscured is nice, but not enough. It should be clear what the limits on this are, e.g.: (a) what is definitely -- has to be -- exposed? (b) what does your system do to help? Is encryption automatic? (If so: Can we trust no decryption will be done?) Is encryption the responsibility of the user? (c) etc. ...

Finally, or rather, in the first place, it should be made clear exactly what data is collected.

Despite these issues, since the user has the option to run the system themselves, these things primarily apply when using OSS Vizier when the service is provided by someone.

**Overall Review:**

- A system to run BBO/HPO algorithms in a distributed fashion is useful and a welcome addition.
- Having this as a standalone OSS tool is great.
- Code seems to be of good quality and has tests included, this helps working with it and trusting it.
- Limitations and generality are somewhat unclear, which leaves questions about broadness of applicability and impact. Details under: "Potential Impact On The Field Of AutoML" and "Technical Quality And Correctness".
- The focus of OSS Vizier is discussed as being on HPO, but general BBO is also discussed. This makes the focus somewhat unclear. Details under: "Technical Quality And Correctness Rating".
- The many tools/frameworks/libraries related to in the context of OSS Vizier makes it confusing how OSS Vizier relates to them. Details under: "Technical Quality And Correctness Rating".
- The privacy issues related to data collection by the service provider are discussed, but in somewhat limited detail. Details under: "Ethics Details".

**Potential Impact On The Field Of Automl:**

A framework for distributed BBO/HPO is certainly of interest to the AutoML community. However, although applications definitely exist, the number of applications where distributed computing is needed (rather than just an option/nicety or simply overkill) in this context does not seem especially large to me, limiting the scope of the impact.

The presented APIs appear to be reasonably useable, and should not be a major limiting factor on the impact. High-level limitations of the APIs are discussed, but clarity (particularly for the algorithm API) could be better and is hard to assess in the short review time. This makes it somewhat unclear how generally applicable OSS Vizier is, and how general it's impact potential is. Even so, it appears to be broadly usable, even if the necessity for a distributed approach for all possible applications may not be there, it is still nice that it is possible.

Furthermore, the current algorithm availability is a large weakness. Few algorithms are available in the OSS Vizier repository. In addition, it is not clear how useful the already publicly available algorithms are. E.g., the (only) multi-objective algorithm is NSGA-II. This is a good algorithm in principle, but it is not clear if or how it is adapted to the OSS Vizier context. Since NSGA-II works on real-coded problems, how it is adapted to deal with the mixed-variable spaces considered with OSS Vizier is not only important, but would require additional analysis to determine whether its normal performance still holds after adaptation. Finally, there is no substantiation that the included algorithms are good for HPO, which appears to be the author's primary intended application. Continuing with NSGA-II, this is a good BBO algorithm, but that doesn't automatically make it a good HPO algorithm. Note that in this discussion NSGA-II is an example and that these concerns apply more generally. Although indeed users can add their own algorithms, the question is how easy this is to do in practice (we need a little more than the author's claim of it being easy). I would think adoption (and the impact that follows from it) depends substantially on the - especially initially - available set of algorithms.

Concluding on impact: There is potential for substantial impact in a sub-area of AutoML, but whether this will happen depends on many more factors than the design of the framework, and some of these factors currently remain somewhat vague.

**Reproducibility:**

Despite the clarity issues discussed above, someone else could probably make a reasonably accurate reproduction of OSS Vizier. Of course the public code also helps for this.

As discussed in "Potential Impact On The Field Of AutoML" and "Ethics Details" I the discussion of limitations could be better, as could the discussion of data collection (negative impacts).

The reproducibility list is otherwise filled out reasonably. Note that most parts don't apply (no theoretical results, no experiments, no crowdsourcing/human subjects).

**Review Confidence:**

3: You are fairly confident in your assessment. It is possible that you did not understand some parts of the submission or that you are unfamiliar with some pieces of related work.

**Review Rating:**

4: Marginally above the acceptance threshold (use sparsely)

**Review Summary:**

This work presents a system to run BBO/HPO algorithms in a distributed fashion, which is both useful and a welcome addition. The main contribution is the release of the open source tool OSS Vizier, which provides APIs to optimise using existing public algorithms, and an API to add one's own algorithm implementations (and, presumably, optionally makes those publicly available). The public code seems to be of good quality and has tests included, this helps working with it and trusting it.

Unfortunately, there are also some issues with the current version of the paper. Limitations and generality of the tool are discussed, but in somewhat limited detail, which leaves questions about broadness of applicability (what can it be used for and what not) and impact. Further, while a focus on HPO is stated, discussion of both BBO and HPO leaves some confusion as to the goal.

Confusion is also caused by the many tools/frameworks/libraries related to in the context of OSS Vizier. At the same time, however, these may be of interest to users of those other tools. The privacy issues related to collection of data by the service provider are discussed, but also some cause for concern. However, since users can also set up their on service, it is possible for them to avoid this.

Overall, the paper and the tool have potential to be useful. The presentation clarity could be improved regarding (a) the broadness of applicability, (b) what is the exact relation to other systems. I believe the current version of the paper is not exceptional, but sufficient to be accepted. I want to encourage the authors to make the suggested improvements, and I hope to see the paper published.

**Technical Quality And Correctness:**

Overall, the OSS Vizier API and framework appear to be well thought out, and implemented. Even so, I have some concerns about the clarity (discussed below), and the limitations. The discussion of limitations (or the reverse: areas of applicability) is there on a broad level, but claims of being general would be stronger with strict definitions of limitations. Specific points are:
- Sect. 6.1: "Policy is designed to be a minimal and general-purposed interface" --> How general purpose is it? True generality is clearly hard to achieve, so I think the possibilities and restrictions should be discussed; particularly because these influence which algorithms the user can actually add.
- Sect. 8: "Algorithms may easily be added as policies to OSS Vizier’s collection over time from contributors." --> Related to the above, I seriously question how easy it is to add algorithms. Figuring out how an algorithm can be described (and likely at least partially re-implemented) to fit some generic interface can be quite hard when it was not designed with this interface in mind.

The related work section (2.1) is 'non-comprehensive' as the authors write, and they focus on HPO. For BBO the authors defer to review articles. Additionally, the related work almost exclusively discusses services. This can be OK, but if so you should make it clear from the start that this is your focus.

---

### Official Review · Reviewer_YeY8 · 2022-04-05

**Potential Impact On The Field Of Automl Rating:** 3
**Technical Quality And Correctness Rating:** 3
**Clarity Rating:** 3
**Ethics Rating:** Yes, other reasons (please specify be…

**Summary Of Contributions:**

The paper describes Open Source (OSS) Vizier. Vizier is a blackbox optimization service that has been used extensively internally at Google. The key aspect that sets OSS Vizier apart from existing frameworks/libraries is that it is designed to be deployed in a massive parallel distributed setting, and provides various solutions (communication protocols, faut tolerance, etc.) that enable it to scale, while at the same time staying conceptually simple, providing a high-level pythonic API, both for users (e.g., using BBO to do HPO) and developers (implementing an testing BBO algorithms).

**Clarity:**

The paper is well-written. Nonetheless, not having a software engineering (SE) background / being familiar with Vizier terminology, I found myself more than a little confused in the earlier parts of the paper. While I understand that the paper’s main contributions are of an SE nature, the paper should target a non-SE audience. Therefore, I would strongly suggest explaining the core concepts / terminology (section 4) earlier on, ideally extending this with further examples, to give the reader a better appreciation for the system itself (and the problem it solves) before diving into the technical details.

**Ethics Details (Optional):**

To clarify: I did not find this work unethical. However, the provided system promotes “a massive parallel black box approach to AutoML” (in favor of more whitebox sample efficient approaches). This approach raises various concerns, e.g.,
-	equality (the need for expensive computational resources to do AutoML / research)
-	ecological
that I feel the authors should address in the impact statement.

**Overall Review:**

As listed on https://automl.cc/call-for-papers-on-special-track-for-systems-benchmarks-and-challenges/ a good system’s paper should show that
1.	It is a novel system that has features or application domains that were not available beforehand.
2.	It already has an established user base (shown by stars on github, active commit history by several developers, an active issue tracker, etc.)
3.	It is an open-source software package with an open-source software licence that allows users to easily use and contribute to it.
4.	It achieves excellent performance on the addressed application domains.

(ordered from positive to negative) In this work, the authors…
3) open source a proprietary system.
1) describe a system providing various general features. While many of these are not novel (many other libraries/frameworks support them), the fact that the system supports them all in a way that is not method specific, has the potential to allow AutoML researchers to leverage these features to focus on algorithmics rather than software engineering.
2) describe a system used exclusively(?) for Google internal projects (the Github repo has 15 stars). I am not sure this constitutes an active user base (will Google use the open-sourced version internally?). Also, broader adoption in the autoML  is non-obvious (see impact discussion)
4) do not provide any empirical evidence (see technical concerns).


**Potential Impact On The Field Of Automl:**

OSS Vizier provides a solution for various design problems that AutoML systems face when trying to make most out of the available parallel processing resources. Here, most rely on ad hoc solutions that tend to complicate their implementation and maintenance.
OSS Vizier therefore has the potential of (i) relieve AutoML researcher from these engineering issues, allowing them to focus on algorithm design and (ii) offer a better / more reliable user experience.

However, key for this potential to materialize is “adoption by the community at large”. This is non-trivial and non-obvious at this point (not sure usage within Google can be seen as a proxy). Black box optimization (in the strict sense) is hardly regarded state-of-the-art in the AutoML community, at least in smaller scale settings where sample efficiency matters, which arguably constitutes the majority of the community. Here, speed up techniques, e.g., Successive Halving (also referred to as grey box methods) are essential, and it is unclear to me whether the API can support such methods.

**Reproducibility:**

There were no experiments, so nothing to reproduce.

**Review Confidence:**

4: You are confident in your assessment, but not absolutely certain. It is unlikely, but not impossible, that you did not understand some parts of the submission or that you are unfamiliar with some pieces of related work.

**Review Rating:**

4: Marginally above the acceptance threshold (use sparsely)

**Review Summary:**

This paper open sources a proprietary system that claims great usefulness to the AutoML community.  I acknowledge that the system itself is interesting and that such efforts should be strongly encouraged. Sadly, all arguments the paper provides in support of its claim are of a conceptual nature. In my opinion, even a systems paper should show evidence for its claims, beyond “Google uses it”. It should provide algorithms the community cares about and demonstrate its features empirically. The lack thereof raises concerns about the usefulness of / adoption by the AutoML community and the impact of this work at large. The deciding factor is my decision (weak accept vs. reject) is my believe that the system, despite the lack of evidence, is useful. However, even if accepted, I would strongly encourage the authors to go through the extra effort of providing such evidence, as I feel that doing so would greatly increase the impact of their work.

**Technical Quality And Correctness:**

While the system seems conceptually sound, I find the lack of evidence of algorithmic / empirical nature concerning. I fully understand that Google’s internal algorithms cannot be released, but if the provided APIs are truly as flexible as the authors claim, it should not be difficult to provide some canonical / popular implementations, alongside empirical performance evaluations (speedup, fault tolerance, etc.) at different scales. Lacking such evidence, it is impossible for me to assess whether the claims made in the paper are substantial or just a decent sales pitch.

Minor:
-	In Table 1, the features column of HPBandSter is empty. Why? HPBandSter supports various of these features no?

---

### Official Review · Reviewer_KCX1 · 2022-04-05

**Potential Impact On The Field Of Automl Rating:** 3
**Technical Quality And Correctness Rating:** 3
**Clarity Rating:** 4

**Summary Of Contributions:**

This paper describes and provides an open-source python-based interface for black-box optimization (i.e., OSS Vizier) based on the Google Vizier [1]. Compare with the previous work, it proposes an evolved backend design and several new features (e.g., multi-objective). The technical details are presented in the paper, providing a user-friendly black-box optimization toolbox for everyone who tends to solve a black-box optimization problem.

[1] Golovin, D., Solnik, B., Moitra, S., Kochanski, G., Karro, J., & Sculley, D. Google vizier: A service for black-box optimization. In Proceedings of the 23rd ACM SIGKDD International Conference on Knowledge Discovery and Data Mining.

**Clarity:**

The paper is well written and structured, and the presentation is overall clear.


**Overall Review:**

This paper describes and provides an open-source python-based interface for black-box optimization (i.e., OSS Vizier) based on the Google Vizier [1]. Compare with the previous work, it proposes an evolved backend design and several new features (e.g., multi-objective).  The contribution of this paper is clear, but it lacks performance evaluation.

**Potential Impact On The Field Of Automl:**

Hyper-parameter tuning for a given model is one of the most significant ends and aims of an AutoML system. Although the provided open-source service could be widely used to solve various kinds of optimization problems, it has been utilized mainly for hyper-parameter tuning in Google and is critically acclaimed. The paper illustrates several advanced features compared to other open-source black-box optimization toolboxes, showing the strong functionality. I believe it would benefit the AutoML community in the future.

**Reproducibility:**

The code is given and well organized, and the guideline is written in detail. More importantly, accroding to the authous, it has thousands of monthly users both on the research and production side at Google.


**Review Confidence:**

4: You are confident in your assessment, but not absolutely certain. It is unlikely, but not impossible, that you did not understand some parts of the submission or that you are unfamiliar with some pieces of related work.

**Review Rating:**

4: Marginally above the acceptance threshold (use sparsely)

**Review Summary:**

Overall an interesting paper with clear contributions. Its performance evaluation can be further improved.

**Technical Quality And Correctness:**

The system described in this paper is well-developed. It could be viable for commercial use. However, because this paper lacks performance evaluation, it is difficult to determine whether is there any significant advance compared with [1].

---

### Official Review · Reviewer_n85f · 2022-04-05

**Potential Impact On The Field Of Automl Rating:** 4
**Technical Quality And Correctness Rating:** 2
**Clarity Rating:** 3

**Summary Of Contributions:**

This paper proposes Open Source (OSS) Vizier, a Python implementation of Google Vizier's APIs.

**Clarity:**

The paper is well-written in general, though the illustrations are not as straightforward to understand without reading the text.

**Overall Review:**

Proposing open source versions of existing commercial tools are a very important contribution to the literature, especially when key industry players are involved. That being said, the focus of this work is on (i) the architecture proposed and (ii) how the proposed system represents a contribution with regard to other existing systems. I believe the issues with this paper intersect (i) and (ii). In detail, authors have selected a very limited subset of systems to compare to, and their discussion on pros and cons from these systems is even more limited. As a result, when discussing (i), authors present many existing architectural choices as a contribution of the proposed system, when in fact they are indeed good choices, but far from novel. Overall, I recommend that the paper be reviewed to properly value the existing literature on the field, which is mature and highly relevant.

**Potential Impact On The Field Of Automl:**

Being a system proposed by a key industry player, the likelihood of adoption of this system is significant.

**Reproducibility:**

Being a system paper, no experiments are conducted.

**Review Confidence:**

5: You are absolutely certain about your assessment. You are very familiar with the related work and checked all the details carefully.

**Review Rating:**

5: Accept, good paper

**Review Summary:**

This paper proposes a very important contribution to the literature, but I recommend that the paper be reviewed to properly value the existing literature on the field, which is mature and highly relevant.

**Technical Quality And Correctness:**

The paper describes the proposed system in an appropriate level of detail. However, the comparison to related work is fairly limited. Not only a limited sample of available tools is listed, but the design options that underlie the proposed system are fairly common in the literature but are never referenced accordingly. Importantly, this paper addresses black-box optimization from the perspective of the machine learning community, which inevitably leads to lack of important references and statements that are inaccurate ("The most notable examples are evolutionary algorithms, which have seen a surge in popularity since Vizier (Golovin et al., 2017) was first introduced. Evolutionary algorithms are the preferred method in a latency-sensitive loop for optimizing blackbox functions that can be evaluated quickly.").

---

### Official Review · Reviewer_hq6L · 2022-04-05

**Potential Impact On The Field Of Automl:** N/A
**Potential Impact On The Field Of Automl Rating:** 3
**Technical Quality And Correctness:** N/A
**Technical Quality And Correctness Rating:** 3
**Clarity:** N/A
**Clarity Rating:** 3

**Summary Of Contributions:**

N/A

**Overall Review:**

N/A

**Reproducibility:**

Since this paper did not present any results or algorithm, there is not much to do here for "reproducibility".

Here are a few things that I tried and worked.
- Able to clone and install the package.
- Able to run the server following the instruction
- Able to run `vizier/service/vizier_client_test.py` unit test.

Now I tried to follow the README to set up a Study, and I ran into the following error that I wasn't sure how to fix.
```
E0405 20:53:52.424506 139846349747968 _server.py:445] Exception calling application: 'my_name'
```

There is no documentation besides the README to help me diagnose what caused this problem. Therefore, I am not able to test other features further.


**Review Confidence:**

4: You are confident in your assessment, but not absolutely certain. It is unlikely, but not impossible, that you did not understand some parts of the submission or that you are unfamiliar with some pieces of related work.

**Review Rating:**

5: Accept, good paper

**Review Summary:**

N/A

---

### Meta-Review · Area_Chair_j71w · 2022-05-09

**Recommendation:** Accept
**Confidence:** 5

**Metareview:**



This paper introduces an open source vizier (OSS) based on the google-internal vizier infrastructure. OSS vizier gives an API capable of defining and solving various optimization problems and it is designed to be a distributed system.

The paper was after the revision of 6 reviewers with 4.5 in the average evaluated. At least one of the reviewers he went up from the 2 to a 4 because the clarity of the work improved. This already highlights that for this paper the rating has improved substantially because of the way the paper could be improved. As said, it is now in the range of the conference papers as the paper now has a 4.5 on average. The other marks for the potential impact, the technical quality and the clarity rating are all (except of one which is 2) between the 3 and the 4 rating.

Nevertheless, I have to admit that the open source paper is dependent on whether it will be used or not. Also, I don't understand why the 9-pages main PDF cannot be revised to avoid complications. I think it is a must! Hence, I would make it an accept, if in the 9-pages paper there can be improved according to these issues.

---

### Decision · Program_Chairs · 2022-05-13

Accept